# Yeast Kinesin-5 Motor Protein CIN8 Promotes Accurate Chromosome Segregation

**DOI:** 10.3390/cells11142144

**Published:** 2022-07-07

**Authors:** Delaney Sherwin, Abigail Huetteman, Yanchang Wang

**Affiliations:** 1Department of Biomedical Sciences, College of Medicine, Florida State University, 1115 West Call Street, Tallahassee, FL 32306-4300, USA; delaney.sherwin@med.fsu.edu; 2Medical Student, College of Medicine, Florida State University, 1115 West Call Street, Tallahassee, FL 32306-4300, USA; ath16b@med.fsu.edu

**Keywords:** kinesin-5 motor proteins, Cin8, Kip1, tension on chromosomes, tension checkpoint

## Abstract

Accurate chromosome segregation depends on bipolar chromosome–microtubule attachment and tension generation on chromosomes. Incorrect chromosome attachment results in chromosome missegregation, which contributes to genome instability. The kinetochore is a protein complex that localizes at the centromere region of a chromosome and mediates chromosome–microtubule interaction. Incorrect chromosome attachment leads to checkpoint activation to prevent anaphase onset. Kinetochore detachment activates the spindle assembly checkpoint (SAC), while tensionless kinetochore attachment relies on both the SAC and tension checkpoint. In budding yeast *Saccharomyces cerevisiae*, kinesin-5 motor proteins Cin8 and Kip1 are needed to separate spindle pole bodies for spindle assembly, and deletion of *CIN8* causes lethality in the absence of SAC. To study the function of Cin8 and Kip1 in chromosome segregation, we constructed an auxin-inducible degron (AID) mutant, *cin8-AID*. With this conditional mutant, we first confirmed that *cin8-AID kip1∆* double mutants were lethal when Cin8 is depleted in the presence of auxin. These cells arrested in metaphase with unseparated spindle pole bodies and kinetochores. We further showed that the absence of either the SAC or tension checkpoint was sufficient to abolish the cell-cycle delay in *cin8-AID* mutants, causing chromosome missegregation and viability loss. The tension checkpoint-dependent phenotype in cells with depleted Cin8 suggests the presence of tensionless chromosome attachment. We speculate that the failed spindle pole body separation in *cin8* mutants could increase the chance of tensionless syntelic chromosome attachments, which depends on functional tension checkpoint for survival.

## 1. Introduction

During mitosis, chromosomes are attached to microtubules originating from opposite spindle poles to establish bipolar attachment, which ensures equal segregation of sister chromatids into each daughter cell. Bipolar attachment generates tension on chromosomes, and in response to incorrect chromosome attachments or lack of tension, the spindle assembly checkpoint (SAC) becomes activated to prevent anaphase initiation [1,2]. In budding yeast, the components of the SAC include Mad1, Mad2, Mad3, Bub1, Bub3, and protein kinase Mps1, which are conserved in all eukaryotes and localize to unattached kinetochores [3,4,5]. SAC activation arrests cells in metaphase to prevent sister chromatid segregation and allows time for correction of improper chromosome attachment. The establishment of chromosome bipolar attachment generates tension that silences the SAC and allows cells to proceed into anaphase. Defects in the SAC result in chromosome missegregation or aneuploidy, a hallmark for most solid tumors and the cause of Trisomy 21 [6,7].

The establishment of chromosome bipolar attachment depends on two intertwined events. First, the separation of spindle pole bodies (SPBs) by motor proteins ensures bi-oriented microtubules from opposite spindle poles, and this is required for chromosome bipolar attachment [8]. Second, kinetochores must be oriented so that only microtubules originating from opposite spindle poles can bind, thereby establishing bipolar attachment. However, a previous study using un-replicated dicentric chromosomes in budding yeast suggests that tension alone promotes bi-oriented microtubule attachment, regardless of kinetochore orientation [9]. One explanation is that an intrinsic mechanism corrects improper attachment to ensure chromosome bipolar attachment in the absence of sister kinetochore orientation. Additionally, the tension generated between sister kinetochores by bipolar microtubule attachment further stabilizes the kinetochore–microtubule interaction.

In budding yeast, tension generation is monitored by a specific pathway called the tension checkpoint, which consists of the SAC, Aurora B kinase Ipl1, and centromere-associated protein Sgo1 [10,11]. Previous results in budding yeast support a model that Ipl1 kinase prevents SAC silencing in the presence of tensionless attachment by phosphorylating kinetochore protein Dam1 [12,13,14]. Both *ipl1* and phospho-deficient *dam1-3A* mutantd show competent checkpoint function in response to detached chromosomes resulting from microtubule depolymerization but fail to arrest at metaphase during tensionless attachments [10,13]. Similarly, syntelic attachment occurs when sister kinetochores are attached by microtubules from one SPB, giving way to a tensionless attachment, but *ipl1* and *dam1-3A* mutants abolish the anaphase entry delay caused by this tensionless attachment [13]. Ipl1 kinase has also been shown to destabilize kinetochore–microtubule interactions to facilitate corrections [15,16]. This feature raises the possibility that the tension checkpoint pathway consequently activates the SAC by generating detached kinetochores [17,18]. Therefore, Ipl1 kinase plays a central role in coordinating the correction of tensionless attachment and checkpoint control.

Aside from kinetochore proteins, microtubule-associated motor proteins also play a role in establishing and maintaining chromosome bipolar attachment. In budding yeast, Cik1 forms a complex with Kar3, a minus-end-directed kinesin-14 motor protein [19]. The Cik1/Kar3 complex associates with chromosomes and facilitates their poleward movement at the start of microtubule depolymerization [20,21]. Loss of function of the Cik1/Kar3 complex results in syntelic chromosome attachments, as evidenced by chromosome missegregation when combined with tension checkpoint mutants [22].

In budding yeast, kinesin-5 motor proteins Cin8 and Kip1 are involved in spindle assembly and kinetochore attachment [23,24]. Cin8 shows bi-directional motility [25]. The minus-end-directed motility of Cin8 enables its clustering near spindle poles, where it captures microtubules from the opposite spindle poles and mediates their antiparallel sliding. Therefore, this activity is critical for microtubule crosslinking and the initial separation of the spindle poles for spindle assembly [26]. Consistently, temperature-sensitive *cin8-3* mutant cells exhibit impaired separation of spindle poles, and this phenotype is more pronounced in *cin8-3 kip1* double mutants at high temperatures [23,27]. Cin8 and Kip1 also promote the disassembly of longer kinetochore microtubules to facilitate chromosome congression [28]. Moreover, Kip1 is involved in the spindle midzone localization of the chromosomal passenger complex for spindle disassembly in late anaphase [29]. Interestingly, *cin8∆*, but not *kip1∆*, is synthetically lethal with SAC mutants, indicating that Cin8 plays a major role in accurate chromosome segregation [27,30]. To that point, a recent study showed that Cin8 localizes to the kinetochore during metaphase, directly interacts with the Dam1 kinetochore complex, and is required to generate tension at kinetochore protein Ndc80 [31]. In another study, a significant decrease in tension during metaphase was observed in cells lacking Cin8 [32].

Given these known functions of Cin8 and Kip1, we speculated that these motor proteins might be required for tension generation and chromosome segregation. To further understand their function, we constructed a conditional *cin8-AID* (auxin-inducible degron) mutant to show that cells lacking Cin8 and Kip1 present defective separation of SPBs and sister kinetochores. This arrest was SAC-dependent, and interestingly, combination with tension checkpoint mutants caused an increased rate of chromosome missegregation and viability loss. We speculate that the failure of SPB separation increases the chance of tensionless syntelic chromosome attachment, which depends on functional tension checkpoint for survival. In all, our research has uncovered kinesin-5 motor proteins Cin8 and Kip1 as key players in tension generation and accurate chromosome segregation.

## 2. Methods

### 2.1. Yeast Strains, Growth, and Media

The relevant genotypes and sources of the yeast strains used in this study are listed in Appendix A. All the strains listed are isogenic to Y300, a W303 derivative, and they were constructed by tetrad dissection. Yeast cell growth conditions and synchronization were performed as described previously [33].

### 2.2. Generating cin8-AID Conditional Mutants

The primers used to construct the *cin8-AID* strain were designed as instructed [34]. The AID template plasmid used for PCR reaction in this research was modified with three Flag tags in the AID cassette (pXZ830 *IAA17-3×Flag::HIS3*, a gift from the Xiaolan Zhao Lab at the Sloan Kettering Institute, New York City, NY, USA) [35]. The PCR fragments were transformed into an *ADH1-osTir1-9×myc-URA3* (X3338-8D) strain. Colonies were selected from HIS dropout plates, and their growth on auxin plates, as well as Cin8-AID protein level after 500 µM auxin treatment, was examined via Western blotting. The resulting *cin8-AID::HIS3 ADH1-osTir1-9myc-URA3* strain was used in subsequent crosses.

### 2.3. Budding Index

For the indicated time points, samples were taken from the culture and fixed with 10% formaldehyde, at a final concentration of 3.7%. Cells were counted and categorized as single cell, small bud, and large bud based on the existence and size of a daughter cell. A cell was counted as large-budded when the diameter of a daughter cell was greater than half of the diameter of the mother cell. The percentage of large-budded cells out of 100 was plotted.

### 2.4. Plating Efficiency/Viability

A small volume of cells was 10-fold diluted and then spread onto a YPD plate. After incubation at 30 °C overnight, cell viability was examined under a microscope and categorized as viable and nonviable. Cells that formed mini-colonies were counted as viable, while nonviable cells were determined as single dead cells or a small cluster of sick cells. For each strain, more than 300 cells were counted to obtain the percentage of viable cells.

### 2.5. Western Blotting

Yeast cells (1 mL) were collected by centrifugation, and the cell pellets were resuspended in 200 μL of 0.1 M NaOH. After incubation at room temperature for 5 min, the sample was centrifuged, and the pellets were resuspended in 100 μL 1 × SDS protein loading buffer. The protein samples were then boiled for 5 min and resolved by 8% SDS–PAGE. After probing for anti-myc (9E10, Covance Research Products, Inc., Denver, PA, USA), anti-Flag (Sigma-Aldrich, St. Louis, MO, USA), and anti-Pgk1 antibodies (Molecular Probes, Eugene, OR, USA), followed by horseradish peroxidase-conjugated secondary antibody (Cell Signaling Technology, Danvers, MA, USA), the protein levels were detected with enhanced chemiluminescence (PerkinElmer, Waltham, MA, USA). A Bio-Rad (Hercules, CA, USA) ChemiDoc imaging system was used to image blots.

### 2.6. Quantification of Pds1 Protein Levels

The relative levels of Pds1 protein for Western blots were quantified with ImageJ. The intensity of Pds1 and Pgk1 protein bands for each time point was measured in arbitrary units. The ratio of Pds1 intensity to Pgk1 was then determined. The time point with the highest ratio was set as 1, and each time point was normalized to this. The resulting numbers were then plotted on a line graph to show the change of relative Pds1 protein levels during the cell cycle.

### 2.7. Cytological Techniques

For fluorescence microscopy, collected yeast cells were fixed with 3.7% formaldehyde for 5 min and then washed once with water. Cells were then resuspended in 1 × PBS (pH 7.2) for the examination of fluorescence signals, using a microscope with a 60× objective (BZ-X800 from Keyence, Itasca, IL, USA). Images were taken with appropriate channels for mCherry, GFP, and brightfield and z-stacks were created with the stack set to 0.2 µm. BZ-X800 software was used to create composites.

### 2.8. Statistical Analysis

The results from fluorescence microscopy and budding index experiments were determined by counting 100 cells for each yeast strain, with three experimental repeats. The results from viability experiments were determined by counting 300 colonies with three experimental repeats. We then performed either a Wilcoxon rank sum test or a non-parametric Kruskal–Wallis one-way ANOVA and determined the *p*-values. The exact test used is indicated in the figure legends. Statistical significance was determined when *p* < 0.05 (*) and is denoted as such.

## 3. Results

### 3.1. Construction of cin8-AID Conditional Mutant

To study the function of kinesin-5 motor proteins Cin8 and Kip1 in chromosome segregation, we generated a conditional *cin8-AID* mutant. When the AID cassette is fused with a protein of interest, this cassette targets the fused protein for degradation in the presence of auxin [34,36]. We constructed the *cin8-AID* mutant by transforming PCR-amplified DNA fragments containing AID-Flag into yeast cells, which was inserted after the *C*-terminus of the endogenous *CIN8* gene in the budding yeast genome via recombination. The Tir1 F-box protein was also introduced into *cin8-AID* yeast strains [35]. The presence of indole-3-acetic acid (IAA/auxin) would cause rapid poly-ubiquitination of Cin8-AID by SCF^Tir1^ E3 ubiquitin ligase and the subsequent degradation by the proteasome [34]. To verify the *cin8-AID* strain, we first examined the degradation of Cin8-AID in the presence of auxin. This AID cassette contains a Flag tag, so the levels of Cin8 were analyzed in asynchronous *cin8-AID* cells after incubation in the presence of 500 μM auxin via Western blotting with anti-Flag antibody. The results showed a decrease in Cin8 protein levels after release into auxin medium for just 20 min, with nearly full clearance of the Cin8 protein after 40 min (Figure 1A).

Budding yeast contains two kinesin-5 motor proteins, Cin8 and Kip1. Cin8 has a predominant function in chromosome attachment because the *cin8∆*, but not the *kip1∆*, mutant shows synthetic lethality with SAC mutants. In addition, *cin8∆* and *kip1∆* mutants are synthetically lethal [30,31,37]. Because Cin8 depletion by AID was expected to cause lethality in *kip1∆* cells, we also constructed *cin8-AID* in a *kip1∆* background and confirmed the clearance of Cin8 protein after auxin addition (Figure 1A). The *cin8-AID kip1∆* grew similarly to wild-type (WT) cells on yeast extract–peptone–dextrose (YPD) plates but failed to grow on a plate supplemented with 500 μM auxin (Figure 1B); such a result is consistent with previous observations [29]. Thus, the synthetic lethality between *cin8-AID* and *kip1∆* in the presence of auxin further confirmed the efficient Cin8-AID degradation induced by auxin. We also observed 49% viability loss in *cin8-AID kip1∆* cells after 6 h of incubation in auxin medium, compared to only a 5% decrease in *cin8-AID* single mutant, which is a statistically significant difference (Figure 1C). Furthermore, we observed an accumulation of large-budded cells in *cin8-AID kip1∆* mutants in the presence of auxin, indicating halted cell-cycle progression (Figure 1D).

### 3.2. The Absence of Both Cin8 and Kip1 Arrests Cells at Metaphase with Unseparated Kinetochore Clusters and SPBs

We first examined the cell-cycle progression in WT, *cin8-AID*, and *cin8-AID kip1∆* cells. After G_1_ release into medium containing 500 μM auxin, WT cells showed normal progression through the cell cycle, as indicated by the increase and decrease of large-budded cells during cell cycle, while *cin8-AID* and *cin8-AID kip1∆* maintained high numbers of large-budded cells at later time points, with the *cin8-AID kip1∆* double mutants showing a stronger phenotype (Figure 2A). We also analyzed Pds1/Securin level in these synchronized cells. Pds1 protein is the anaphase inhibitor in budding yeast, and its degradation by the ubiquitin–proteasome system indicates SAC silencing and anaphase entry [38,39]. In WT cells, Pds1 levels showed an increase after G_1_ release, followed by a decrease, indicating anaphase onset (Figure 2B). The *cin8-AID* single mutants showed normal Pds1 increase after G_1_ release but delayed Pds1 degradation. The Pds1 levels in *cin8-AID kip1∆* cells, however, remained persistent beyond the initial increase after G_1_ release, thus indicating that the absence of both Cin8 and Kip1 prevents anaphase entry (Figure 2B). The levels of Pds1 at each time point relative to the Pgk1 loading control were plotted, and the results further support the increased Pds1 stability in *cin8-AID* and *cin8-AID kip1∆* cells (Figure 2C).

To further determine the underlying mechanism of the anaphase entry delay in *cin8-AID kip1∆* mutant cells, we constructed strains containing GFP-marked kinetochore protein Mtw1 and mCherry-labeled SPB protein Spc110 to visualize kinetochore separation and spindle elongation. WT cells showed normal cell-cycle progression, as indicated by an increase followed by a decrease in large-budded cells after G_1_ release into auxin media. However, many more *cin8-AID* and *cin8-AID kip1∆* cells remained large-budded at later time points (Figure 2D), which is similar to the cell-cycle kinetics shown in Figure 2A. After release into auxin media for 100 min, 95% of WT cells showed normal separation of both Mtw1 kinetochore clusters and SPBs into each daughter cell, and only 2% of cells showed unseparated kinetochores and SPBs (Figure 2E,F). In clear contrast, at the same time point, 91% of *cin8-AID kip1∆* cells showed unseparated kinetochore clusters and SPBs. This number was similar at 120 min, indicating a tight metaphase arrest (Figure 2E,F). This is consistent with the known function of kinesin-5 motor proteins in SPB separation [23,27,40,41]. For *cin8-AID* single mutant cells, 100 min after release from G_1_ arrest, 79% showed metaphase arrest with one Mtw1/SPB cluster, and this number decreased to 73% at 120 min, exhibiting a less severe phenotype than *cin8-AID kip1∆* cells (Figure 2E,F). This result indicates that Kip1 is still able to promote the separation of SPBs and sister kinetochores in some cells when Cin8 is absent, but the absence of both Cin8 and Kip1 blocks kinetochore/SPB separation almost completely.

### 3.3. Cin8-AID Mutants Show Chromosome Missegregation in the Absence of the SAC

The presence of detached chromosomes or the absence of tension on sister kinetochores activates the SAC to prevent anaphase entry [2]. Mad1 is a component of the SAC, and *mad1∆* mutant cells are able to enter anaphase, even in the presence of incorrect chromosome attachment [5]. We reasoned that *mad1∆* would alleviate the metaphase arrest caused by a possible defect in chromosome attachment and/or tension generation in *cin8-AID* mutants, resulting in chromosome missegregation and viability loss. To test this idea, we first examined the growth of *cin8-AID mad1∆* strains in the presence of auxin. As expected, *cin8-AID mad1∆* cells displayed a clear growth defect on plates containing 500 μM auxin compared to either *cin8-AID* or *mad1∆* alone, thus suggesting synthetic lethality (Figure 3A).

To test if the absence of SAC in *cin8-AID mad1∆* cells causes chromosome missegregation, we constructed a *cin8-AID mad1∆* strain containing GFP-marked centromere of chromosome IV (*CEN4*-GFP) and Spc110-mCherry that marks SPBs [42]. We chose to use GFP-marked *CEN4* over Mtw1 to examine chromosome segregation because kinetochore clustering in *S. cerevisiae* during cell division would make it difficult to detect the low frequency of chromosome missegregation, if any, using strains with Mtw1-GFP [43]. Strikingly, the loss of the SAC in *cin8-AID mad1∆* mutant cells resulted in a statistically significant increase (23%) in chromosome missegregation after incubation in auxin medium for 3 h, as evidenced by the presence of a cell body with two *CEN4*-GFP dots and one SPB (Figure 3B,C). There is, however, partial spindle pole body separation in *cin8-AID mad1∆* cells, thus suggesting some spindle function in these mutants. This mirrors a previous study that showed that SAC was required to prevent premature SPB separation when spindle function was impaired [27]. Consistently, *cin8-AID mad1∆* cells showed much less viability (13%) compared to *cin8-AID* single mutant (81%) or *mad1∆* single mutant (97%) after auxin exposure for 3 h (Figure 3D).

We further examined chromosome segregation in synchronized *cin8-AID mad1∆* cells treated with auxin. WT and *mad1∆* single mutants demonstrated typical cell-cycle progression, while *cin8-AID* cells showed prolonged presence of large-budded cells, suggesting a cell-cycle delay. However, the delay was partially suppressed by *mad1∆* (Figure 3E). After G_1_ release into auxin medium for 105 min to follow chromosome segregation, the majority of *cin8-AID* single mutant cells showed unseparated *CEN4*-GFP dots (73%) (Figure 3F,G). However, 19% of *cin8-AID mad1∆* cells showed chromosome missegregation at 105 min, as evidenced by the presence of two *CEN4*-GFP dots and one SPB in one single cell body. The difference between *cin8-AID mad1∆* cells and *cin8-AID* cells is, therefore, statistically significant (Figure 3F,G). We also acknowledge that *cin8-AID* mutant cells classified with normal chromosome segregation often do not achieve full SPB separation as in WT cells, and this is likely due to loss of motor protein-dependent spindle elongation. In any case, these results indicate a chromosome attachment defect in Cin8-depleted cells that is dependent on the SAC for survival.

### 3.4. Cin8 and Kip1 Work Cooperatively to Promote Faithful Chromosome Segregation

We showed that the loss of the SAC suppressed the metaphase arrest in *cin8-AID* cells, as evidenced by the budding index. Compared to *cin8-AID*, *cin8-AID kip1∆* mutant cells exhibit a more pronounced defect in kinetochore/SPB separation (Figure 2). To test if loss of the SAC also abolishes the cell-cycle arrest of *cin8-AID kip1∆* we first examined cell-cycle progression in synchronized *cin8-AID kip1∆* cells, both with and without SAC, in the presence of auxin. We found that the loss of the SAC in *cin8-AID kip1∆ mad1∆* cells resulted in a decrease in large buds at later time points, which is indicative of cell-cycle progression (Figure 4A). To analyze anaphase entry, we examined Pds1 levels in these cells after G_1_ release into auxin media. The Western blot results indicated normal Pds1 degradation in WT and *mad1∆* cells and persistent Pds1 levels in *cin8-AID kip1∆* cells. However, in *cin8-AID kip1∆ mad1∆* cells, the Pds1 levels were high at 60 min after G_1_ release, but an obvious decrease was detected after that time point (Figure 4B). The kinetics of Pds1 degradation in these cells is reflected by the quantification of relative Pds1 levels, as this indicates an almost complete abolishment of Pds1 accumulation in *cin8-AID kip1∆* cells by *mad1∆* (Figure 4C). Taken together, these results support the conclusion that *mad1∆* abolishes the metaphase arrest in *cin8-AID kip1∆* mutants and allows for anaphase entry, which is consistent with a previous report [27].

We further examined chromosome segregation in *cin8-AID kip1∆* cells in the absence of the SAC. In contrast to the *cin8-AID* single mutant, which showed partial metaphase arrest, almost all *cin8-AID kip1∆* cells arrested at metaphase with no *CEN4*-GFP segregation after incubation of asynchronous cells in the presence of auxin for 3 h (Figure 4D,E). This indicates the redundant role of Cin8 and Kip1 in spindle formation and further chromosome segregation. Interestingly, after 180 min in auxin, *cin8-AID kip1∆* cells presented with Spc110-mCherry and *CEN4*-GFP signals displaced from the bud neck (Figure 4D). A previous report showed that loss of Cin8 caused unbalanced number of astral microtubules from the two SPBs, resulting in a nuclear positioning defect. Cells lacking Kip1 showed a less dramatic but similar phenotype [44]. Moreover, the loss of the SAC in *cin8-AID kip1∆ mad1∆* cells caused significant *CEN4*-GFP missegregation (27%) (Figure 4D,E), compared to 23% for *cin8-AID mad1∆* cells after 3 h of auxin treatment (Figure 3B,C). In addition, only 9% of *cin8-AID kip1∆ mad1∆* cells were viable after incubation in the presence of auxin for 3 h, but 84% of *cin8-AID kip1∆* cells were viable (Figure 4F). Therefore, complete metaphase arrest in *cin8-AID kip1∆* cells can also be abolished by SAC mutant *mad1∆*, resulting in significant chromosome missegregation and viability loss.

### 3.5. The Absence of Cin8 Leads to Tensionless Chromosome Attachment

Previous studies indicate that Cin8 likely promotes tension generation at sister kinetochores [31,32]. In the absence of tension, the phosphorylation of kinetochore protein Dam1 by Ipl1 kinase prevents the cell from premature anaphase entry. Mutation of three Ipl1 phosphorylation sites on Dam1 (*S257A*, *S265A*, and *S292A*) in *dam1-3A* allows the cell to prematurely enter anaphase in the presence of tensionless attachments [12,13]. It is possible that the anaphase entry delay displayed in *cin8-AID* is triggered by tensionless attachment and requires Dam1 phosphorylation. To test this idea, we first examined the growth of *cin8-AID dam1-3A* cells on auxin plates. The *cin8-AID dam1-3A* showed a severe growth defect compared to either *cin8-AID* or *dam1-3A* single mutant (Figure 5A). We then followed cell-cycle progression in *cin8-AID* and *cin8-AID dam1-3A* after G_1_ release into auxin media. In *cin8-AID* cells, we observed a relatively persistent high number of large-budded cells at the later time points. However, *cin8-AID dam1-3A* cells showed an obvious decrease in large-budded cells, indicating anaphase entry (Figure 5B). In examining Pds1 levels in these cells, we noted that *cin8-AID* cells showed clearly stabilized Pds1 levels compared to WT cells, but the Pds1 levels dropped in *cin8-AID dam1-3A* cells at later time points during the cell cycle (Figure 5C,D). Therefore, phospho-deficient Dam1 abolishes the anaphase entry delay in *cin8-AID* cells, at least partially based on the Pds1 protein level and budding index. Because *dam1-3A* mutant cells show checkpoint defect in response to tensionless chromosome attachment but not chromosome detachment [13], this result indicates the presence of tensionless attachments in cells lacking Cin8.

If the tensionless attachment in *cin8* mutants relies on Dam1 phosphorylation to prevent anaphase entry, then abolishment of this phosphorylation will result in chromosome missegregation. To test this idea, we visualized chromosome segregation in *cin8-AID dam1-3A* cells with *CEN4*-GFP and Spc110-mCherry. After incubation in auxin for 3 h, 43% of *cin8-AID* cells displayed a single *CEN4*-GFP dot, and no *CEN4*-GFP missegregation was observed (Figure 6A,B). In clear contrast, 18% of *cin8-AID dam1-3A* cells showed *CEN4*-GFP missegregation after auxin exposure for 3 h (Figure 6A,B). Not surprisingly, *cin8-AID dam1-3A* cells exhibited greater viability loss (19% viable cells) than *cin8-AID* single mutant (82% viable cells) after 3 h of incubation in the presence of auxin (Figure 6C).

We also analyzed the genetic interaction of *cin8-AID* with temperature-sensitive mutant *ipl1-321*, as functional Ipl1 is also required for metaphase arrest in response to tensionless attachment [10]. The *cin8-AID ipl1-321* cells grew poorly on auxin plates, even at the permissive temperature 25 °C, compared to each single mutant (Figure 7A). We suspected that *ipl1-321* would also suppress the anaphase entry delay in *cin8-AID* cells. We previously showed the loss of tension checkpoint function of *ipl1-321* at 25 °C [13,22]. After following the cell-cycle progression in these strains at 25 °C, a clear alleviation of the accumulation of large-budded cells in *cin8-AID* by *ipl1-321* was observed (Figure 7B). Strikingly, a nearly complete suppression of Pds1 stabilization in *cin8-AID ipl1-321* cells was also observed (Figure 7C,D). Therefore, in combination with our results from *dam1-3A* (Figure 6), we conclude that a defective tension checkpoint in cells with depleted Cin8 causes premature anaphase entry and viability loss. Taken together, these results support the role of Cin8 in tension generation, which is a prerequisite for chromosome segregation. It is likely that the delayed SPB separation in *cin8* mutants increases the chance of tensionless syntelic attachment, which depends on the tension checkpoint to prevent anaphase entry and chromosome missegregation (Figure 7E).

## 4. Discussion

Establishment of chromosome bipolar attachment is essential for the faithful segregation of sister chromatids into two daughter cells. The SAC monitors the mistakes in chromosome–microtubule attachment and the lack of tension at kinetochores to prevent anaphase onset. Cin8 and Kip1 are kinesin-5 proteins responsible for mitotic spindle assembly and chromosome segregation [23,45]. The synthetic lethality between *cin8∆* and SAC mutants indicates the essential role of Cin8 in proper chromosome–microtubule attachment, which could be secondary to the function of Cin8 in spindle assembly [30]. However, the precise function of Cin8/Kip1 in accurate chromosome segregation remains obscure. Here, we present evidence showing the critical role of Cin8/Kip1 in tension generation by using a conditional *cin8-AID* mutant. The phospho-deficient *dam1-3A* mutant shows a checkpoint defect in response to tensionless syntelic attachment, in which sister kinetochores are attached by spindle microtubules from the same spindle pole, but this mutant shows the competent checkpoint in response to chromosome detachment [13]. Interestingly, we found that *dam1-3A* partially abolished the anaphase entry delay in cells lacking Cin8, causing chromosome missegregation and viability loss. Consistently, mutation in Ipl1, the kinase for Dam1, also alleviates the anaphase entry delay in cells lacking Cin8. These results suggest the presence of tensionless chromosome attachment in *cin8* mutant cells, which depends on Ipl1-mediated Dam1 phosphorylation to delay anaphase entry. However, we cannot exclude the possibility that both detached and tensionless chromosome attachments are present in *cin8* or *cin8 kip1* mutant cells.

Recent evidence shows metaphase kinetochore localization of Cin8, and that the Ndc80 and Dam1 kinetochore subcomplexes are required for this localization [31]. In contrast to other kinetochore subcomplexes, the Dam1 complex first associates with microtubules, and its subsequent interaction with the Ndc80 complex secures kinetochore–microtubule interaction [46,47]. One possibility is that the Cin8 motor protein moves the Dam1 complex toward the microtubule plus end to facilitate its interaction with the Ndc80 complex. If that is the case, a decrease in association of the Dam1 complex with the kinetochore is expected. Indeed, this decrease was detected in *cik1* and *kar3* motor mutants, wherein the frequency of syntelic attachment is increased [13,48]. However, the relationship between the decreased kinetochore association of the Dam1 complex and syntelic attachment remains unclear. In addition, Cin8 recruits protein phosphatase PP1 to the kinetochore [31], but the role of Cin8–PP1 in chromosome attachment remains unclear.

We found that the depletion of both Cin8 and Kip1 led to the defective assembly of the metaphase spindle, as indicated by the unseparated SPBs. This observation is consistent with previous observations [23,27]. In addition, anaphase spindles that do elongate in *cin8 kip1* cells have been described as short, and they will break prematurely [29]. One open question is whether the spindle defect contributes to the tensionless attachment in *cin8 kip1* mutants. One possible consequence of the closely located SPBs is that the microtubules from two spindle poles originated from almost the same direction, thus likely increasing the chance of syntelic attachment (Figure 7E). This could explain synthetic lethality between *cin8-AID* and tension checkpoint mutants *ipl1-321* and *dam1-3A*, as well as the chromosome missegregation in *cin8 dam1-3A* cells. Indeed, a previous study suggested that SPB separation promotes chromosome bipolar attachment and, therefore, faithful chromosome segregation [8], but further experiments are required to confirm this possibility in *cin8 kip1* mutants. It is also possible that Cin8 and Kip1 motor proteins stabilize bipolar attachment and/or destabilize syntelic attachment in a SPB separation-independent manner.

We previously showed that partially the elongated spindle structure in S-phase cells compromises kinetochore–microtubule attachment [40]. Therefore, the tight regulation of spindle length is critical for the efficient establishment of correct chromosome attachment. First, an elongated spindle ensures bipolar attachment due to the opposite orientation of microtubules from the two spindle poles but decreases the capability of kinetochore capture due to the longer distance between kinetochores and SPBs. In contrast, a short spindle facilitates kinetochore capture but increases the frequency of syntelic attachment. Our results suggest the important role of different motor proteins in the regulation of spindle length and chromosome segregation.

## Figures and Tables

**Figure 1 cells-11-02144-f001:**
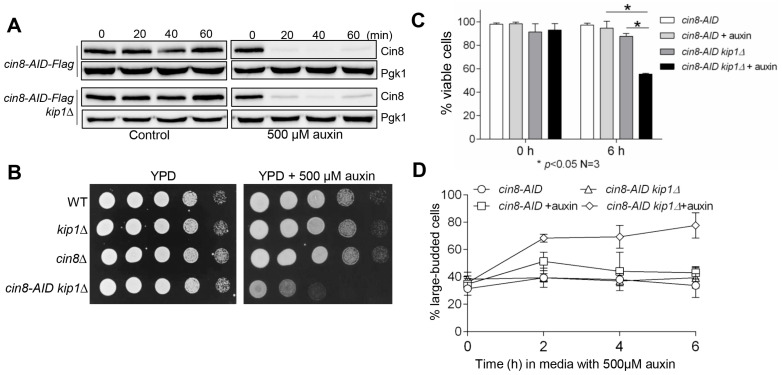
The construction of *cin8-AID* strains. (**A**) Cin8-AID is degraded in the presence of auxin. The *cin8-AID-3×Flag* (3946-2-3, referred to as *cin8-AID* hereafter) and *cin8-AID kip1∆* (3750-3-2) cells were grown in YPD medium to mid-log phase at 30 °C. Auxin (500 μM) was added into half of the cell cultures, and cells were harvested every 20 min. Cin8-AID protein levels were detected after Western blotting with anti-Flag antibody. Pgk1, loading control. (**B**) *cin8-AID kip1∆* double mutants grow poorly in the presence of auxin. Saturated cells of wild-type (WT), *kip1∆* (YBL063W), *cin8∆* (YEL061C), and *cin8-AID kip1∆* (3750-3-2) were 10-fold serially diluted onto YPD plates, both with and without 500 μM auxin. Growth was analyzed after a 2-day incubation at 30 °C. (**C**) The viability of *cin8-AID* and *cin8-AID kip1∆* cells after growth in auxin. The *cin8-AID* (3946-2-3) and *cin8-AID kip1∆* (3750-3-2) cells were grown in YPD medium at 30 °C to mid-log phase, and then 500 μM auxin was added into the cultures. Cells were collected at 0 and 6 h and spread onto YPD plates to examine plating efficiency. Cells that formed mini-colonies after overnight incubation at 30 °C were counted as viable. At least 300 cells were counted for each strain for the percentage of viable cells. The experiment was repeated three times, and statistical significance determined by * *p* < 0.05, using Kruskal–Wallis one-way ANOVA. (**D**) The accumulation of large-budded cells. Cells with the indicated genotypes were grown in the presence of auxin and collected at 2 h intervals to count budding index. A cell was counted as large-budded when the diameter of a daughter cell was greater than half of the diameter of the mother cell. Here shows the average percentage of large-budded cells from three independent experiments.

**Figure 2 cells-11-02144-f002:**
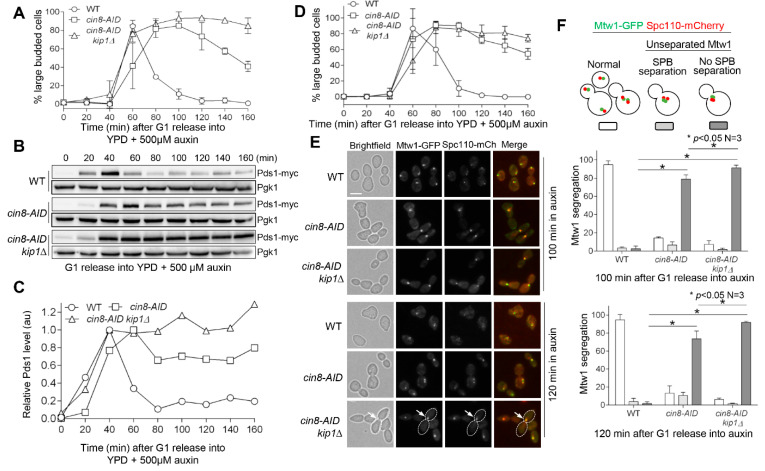
The *cin8-AID kip1∆* mutants show defects in cell cycle and spindle formation. (**A**) Budding index. G_1_-arrested WT (JBY649), *cin8-AID* (3946-2-3), and *cin8-AID kip1∆* (3889-3-1) cells with Pds1-18myc were released into 30 °C YPD medium containing 500 μM auxin. The α-factor was added back after 40 min release to block the following cell cycle. Samples were collected every 20 min to count the budding index. Here shows the average of large-budded cells after G_1_ release from three independent experiments. (**B**) Accumulation of anaphase inhibitor Pds1 in *cin8-AID* mutants treated with auxin. The cells in (**A**) were collected every 20 min to prepare protein samples. Western blotting was performed with anti-myc antibody. Pgk1, loading control. (**C**) Quantification of Pds1 levels. The relative Pds1 levels from the Western blotting results in (**B**) were plotted. Quantification of Pds1 levels is described in the Methods section. (**D**) *cin8-AID* and *cin8-AID kip1∆* cells show mitotic defects. G_1_-arrested WT (4232-6-2), *cin8-AID* (4232-1-2), and *cin8-AID kip1∆* (4232-7-2) cells with Mtw1-GFP and Spc110-mCherry were released into YPD, at 30 °C, containing 500 μM auxin. The α-factor was added back after 40 min to block the following cell cycle. Samples were collected every 20 min for the budding index (*n* = 3). (**E**) *cin8-AID* and *cin8-AID kip1∆* cells display compromised kinetochore or SPB separation in the presence of auxin. The cells in (**D**) were collected and imaged at 100 and 120 min after G_1_ release to visualize the separation of kinetochore clusters (Mtw1-GFP) and SPBs (Spc110-mCherry). White arrows indicate unseparated Mtw1-GFP clusters and SPBs. Scale bar, 5 μm. The images are representative of three experimental repeats. (**F**) Quantification of kinetochore/SPB separation. Kinetochore/SPB separation phenotype was categorized as normal, separated SPB without Mtw1 cluster separation, or unseparated SPB and Mtw1 cluster. The experiment was repeated three times, and statistical significance was determined by * *p* < 0.05, using Kruskal–Wallis one-way ANOVA.

**Figure 3 cells-11-02144-f003:**
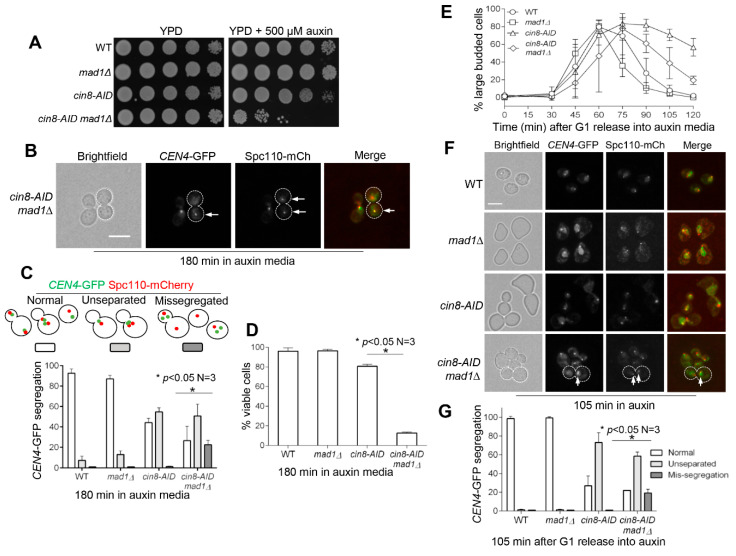
In absence of the SAC, *cin8* mutants lose viability. (**A**) Elimination of the SAC causes growth defect in *cin8* mutant cells. WT, *mad1∆* (4327-9-2), *cin8-AID* (3946-2-3), and *cin8-AID mad1∆* (4236-1-3) cells were 10-fold serially diluted onto YPD plates, both with and without 500 μM auxin. Growth was analyzed after a 2-day incubation at 30 °C. (**B**) Elimination of the SAC results in chromosome missegregation in *cin8* mutants. WT (4244-3-4), *mad1∆* (4327-9-2), *cin8-AID* (4244-2-1), and *cin8-AID mad1∆* (4236-1-3) cells containing *CEN4*-GFP (GFP-marked centromere of chromosome IV) and Spc110-mCherry (mCherry-marked SPB protein Spc110) were grown to mid-log phase in YPD medium at 30 °C. Auxin was added at 500 μM, and pictures were taken at 0 and 180 min. Images show chromosome missegregation in a *cin8-AID mad1∆* cell after incubation in auxin media for 180 min. White arrows represent the SPB and *CEN4*-GFP in this cell. Scale bar, 5 μm. (**C**) Quantification of SPB and chromosome segregation defects. *CEN4*-GFP and SPB segregation at 180 min was counted for the cells in (**B**). *CEN4*-GFP and SPB segregation was categorized as normal, unseparated, or missegregated. Unseparated represents the absence of two clear *CEN4*-GFP and SPB dots. The experiment was repeated three times, and statistical significance was determined by *p* < 0.05, using Kruskal–Wallis one-way ANOVA. (**D**) Deficient SAC causes viability loss in *cin8* cells. Cells in (**B**) were also collected at 180 min and spread onto YPD plates to count the plating efficiency after incubation at 30 °C overnight. The experiment was repeated three times, and statistical significance determined by a * *p* < 0.05, using the Wilcoxon rank sum test. (**E**) The kinetics of cell-cycle progression in synchronized cells lacking Cin8 and SAC. The same yeast strains listed in (**B**) were used in this experiment. G_1_-arrested cells were released into YPD, at 30 °C, containing 500 μM auxin. Samples were collected every 15 min to count the budding index. The α-factor was added back after a 40 min release to block the following cell cycle; *n* = 3. (**F**) Chromosome missegregation in synchronized cells lacking Cin8 and SAC. The same cells were also collected at 105 min after G_1_ release for imaging. White arrows represent SPB and *CEN4*-GFP in a cell showing chromosome missegregation. Scale bar, 5 μm. The pictures are representative of three experimental repeats. (**G**) Quantification of chromosome missegregation in synchronized cells lacking Cin8 and SAC. Segregation of *CEN4*-GFP and SPB at 105 min after G_1_ release was examined. The experiment was repeated three times and statistical significance determined by * *p* < 0.05, using Kruskal–Wallis one-way ANOVA.

**Figure 4 cells-11-02144-f004:**
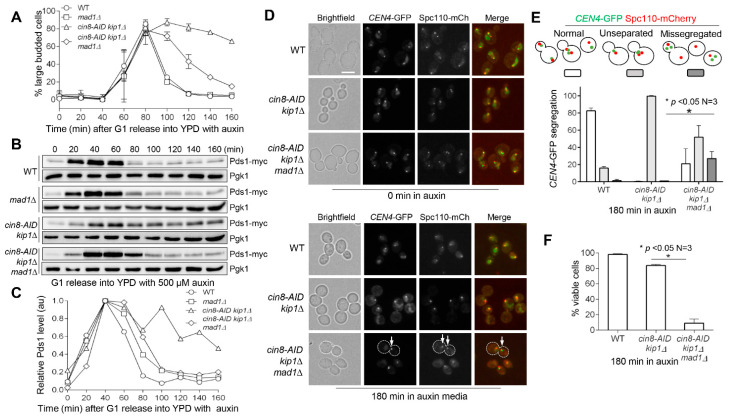
Elimination of the SAC suppresses metaphase arrest of *cin8 kip1* cells. (**A**) *mad1∆* partially suppresses the cell cycle delay in *cin8 kip1* cells. G_1_-arrested WT (JBY649), *mad1∆* (771-4-1), *cin8-AID kip1∆* (3889-3-1), and *cin8-AID kip1∆ mad1∆* (4003-1-3) cells containing Pds1-18myc were released into YPD, at 30 °C, containing 500 μM auxin. The α-factor was added back after 40 min release to block the following cell cycle. Samples were collected every 20 min to count budding index; *n* = 3. (**B**) *mad1∆* alleviates the metaphase arrest in *cin8 kip1* cells. The cells used in (**A**) were collected every 20 min to prepare protein samples. Western blotting was performed with anti-myc antibody to determine the level of Pds1-18myc. Pgk1, loading control. *n* = 3. (**C**) Quantification of Pds1 levels during cell cycle. The relative Pds1 levels of each strain during cell cycle were quantified by using the Western blot results in (**B**). (**D**) Elimination of the SAC results in chromosome missegregation in *cin8 kip1* cells. WT (4244-3-4), *cin8-AID kip1∆* (4260-2-4), and *cin8-AID kip1∆ mad1∆* (4235-9-3) cells containing *CEN4*-GFP and Spc110-mCherry were grown to mid-log phase in YPD medium at 30 °C. Auxin was added at 500 μM. Samples were collected, and pictures were taken at 0 and 180 min. White arrows represent the SPB and *CEN4*-GFP in a cell showing chromosome missegregation. Scale bar, 5 μm. Pictures are representative from three experimental repeats. (**E**) Quantification of chromosome missegregation. Segregation of SPB and *CEN4*-GFP was counted at 180 min. Segregation of *CEN4*-GFP and SPB was categorized as normal, unseparated, or missegregated. Unseparated represents the absence of two clear *CEN4*-GFP and SPB dots. The experiment was repeated three times, and statistical significance was determined by * *p* < 0.05, using Kruskal–Wallis one-way ANOVA. (**F**) Deficient SAC causes viability loss in *cin8 kip1* cells. Cells in (**D**) were also collected at 180 min and spread onto YPD to count the plating efficiency after overnight incubation at 30 °C. The experiment was repeated three times, and statistical significance was determined by * *p* < 0.05, using the Wilcoxon rank sum test.

**Figure 5 cells-11-02144-f005:**
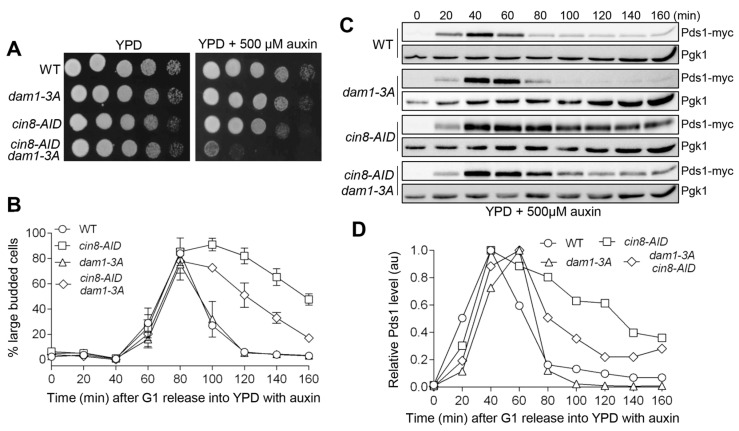
The suppression of the cell-cycle delay in *cin8* cells by tension checkpoint mutant *dam1-3A*. (**A**) *dam1-3A* is synthetically lethal with *cin8-AID* in the presence of auxin. WT (JBY649), *dam1-3A* (2425-7-2), *cin8-AID* (4332-5-4), and *cin8-AID dam1-3A* (4332-13-2) cells with Pds1-18myc were 10-fold serially diluted onto YPD plates, both with and without 500 μM auxin. Growth was analyzed after a 2-day incubation at 30 °C. (**B**) *dam1-3A* partially suppresses the cell-cycle delay in *cin8-AID*. The yeast strains listed in (**A**) were arrested in G_1_ and then released into 30 °C YPD containing 500 μM auxin. The α-factor was added back after 40 min release to block the following cell cycle. Cells were collected every 20 min to count the budding index; *n* = 3. (**C**) *dam1-3A* mutation suppresses the metaphase arrest in *cin8* cells. The yeast cells in (**B**) were collected every 20 min to prepare protein samples. Western blotting was performed with anti-myc antibody to determine Pds1 protein levels. Pgk1, loading control. Pictures are representative of three experimental repeats. (**D**) Quantification of Pds1 levels. The relative Pds1 levels of each strain during cell cycle were quantified by using the Western blotting results from (**C**). Quantification of Pds1 levels is described in the Methods section.

**Figure 6 cells-11-02144-f006:**
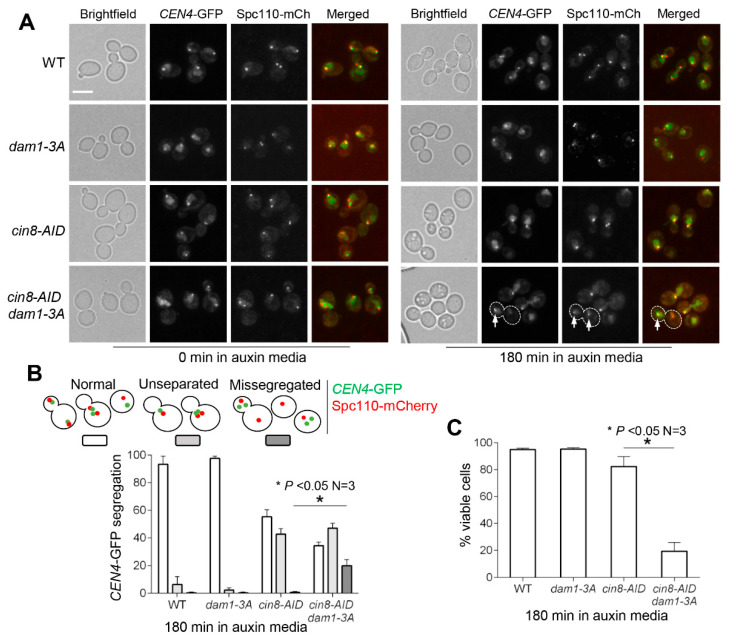
Tension checkpoint mutant *dam1-3A* causes chromosome missegregation in cells with depleted Cin8. (**A**) Images showing chromosome missegregation in *cin8 dam1-3A* mutants. WT (4244-3-4), *cin8-AID* (4244-2-1), *dam1-3A* (4330-7-4), and *cin8-AID dam1-3A* (4244-6-3) cells containing *CEN4*-GFP and Spc110-mCherry were grown to mid-log phase in YPD medium at 30 °C. Auxin was added to a final concentration of 500 μM. Pictures were taken at 0 and 180 min to visualize *CEN4*-GFP and Spc110-mCherry. White arrows represent SPB and *CEN4*-GFP in a cell showing chromosome missegregation. Scale bar, 5 μm. Pictures are representative from three experimental repeats. (**B**) The segregation of *CEN4*-GFP and SPBs. Segregation of *CEN4*-GFP and SPB was categorized as normal, unseparated, or missegregated. This experiment was repeated three times, and statistical significance was determined by * *p* < 0.05, using Kruskal–Wallis one-way ANOVA. (**C**) Viability loss in *cin8-AID dam1-3A* mutants after incubation in the presence of auxin. Cells in (**B**) were also collected at 180 min and spread onto YPD, and the plating efficiency was examined after overnight incubation at 30 °C. The experiment was repeated three times, and statistical significance was determined by * *p* < 0.05, using the Wilcoxon rank sum test.

**Figure 7 cells-11-02144-f007:**
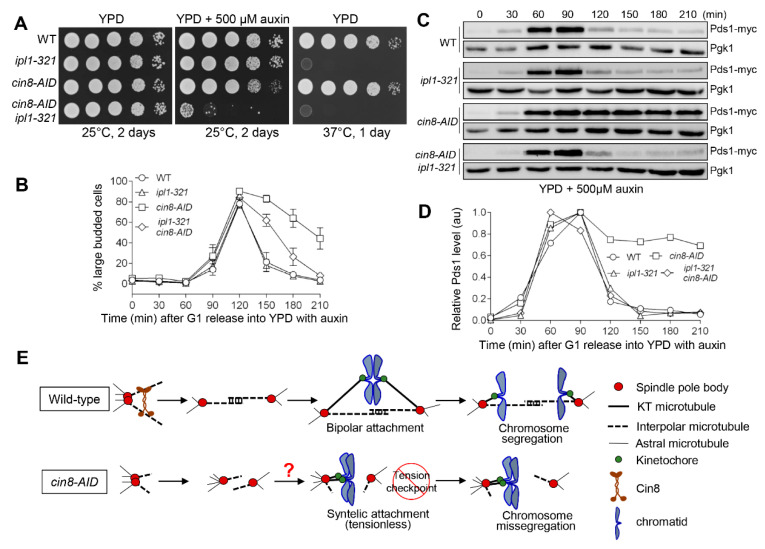
Functional Ipl1 kinase is required for the cell-cycle delay in *cin8* mutants. (**A**) *cin8-AID ipl1-321* double mutants show growth defect on plates containing auxin. Saturated WT (JBY649), *ipl1-321* (2715-6-4)*, cin8-AID* (4332-5-4)*,* and *ipl1-321 cin8-AID* (4338-8-1) cells containing Pds1-18myc were 10-fold serially diluted onto YPD plates, both with and without 500 μM auxin. Growth was analyzed after a 2-day incubation at 25 °C and 1-day incubation at 37 °C. (**B**) *ipl1-321* suppresses the cell cycle delay in *cin8* cells. G_1_-arrested WT (JBY649), *ipl-321* (2715-6-4), *cin8-AID* (4332-5-4), and *cin8-AID ipl1-321* (4333-8-1) cells containing Pds1-18myc were released into YPD at 25 °C containing 500 μM auxin. After 60 min release, α-factor was added back to block the following cell cycle. Samples were collected every 30 min to count budding index (*n* = 3). (**C**) *ipl1-321* mutation suppresses the metaphase arrest in *cin8* cells. The yeast cells in (**B**) were also collected every 30 min to prepare protein samples. Western blotting was performed with anti-myc antibody to determine Pds1 protein levels. Pgk1 was used as the loading control. (**D**) Quantification of Pds1 levels. The relative Pdsl levels were quantified by using the Western blotting results from (**C**). (**E**) A model showing the role of kinesin-5 motor Cin8 in chromosome segregation.

## Data Availability

No new data were created or analyzed in this study. Data sharing is not applicable to this article.

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
