# Peer review of "Yeast Kinesin-5 Motor Protein CIN8 Promotes Accurate Chromosome Segregation"

_cells, 2022, doi:10.3390/cells11142144_

Round 1

Reviewer 1 Report

Sherwin and colleagues use a unique rapid protein depletion system, AID (Auxin inducible degron), to study year Kinesin 5, Cin8 functions in chromosome segregation. Although first half (Figures 1-4) were basically confirmation of previous findings, the use of rapid depletion system provided clean and beautiful results. This work found that mitotic delay induced by Cin8 depletion requires proper dephosphorylation on Dam1 to promote anaphase entry. These findings provided valuable information in the field, and I recommended it for publication in the leading journal, Cells, after addressing few major and minor concerns.

Major concerns:

1)     The title should be “Yeast kinesin-5, Cin8, …”. The study focuses on Cin8 and it is unclear Kip1 contributions.

2)     Although the image of CEN4-GFP in Fig 3B looked good, Figures 3F and 6A were very poor images and difficult to judge the integrity of following quantification.

3)     The use of CEN4-GFP is reasonable but the authors also should perform quantifications of entire kinetochores (such as Mtw1-GFP) and other specific CEN-GFP in addition to CEN4 to support author’s concoctions.

Minor concerns:

4)     In Figure 4D bottom cin8-AID/Kip1KO, why weren’t kinetochore clusters retained at the mid zone (proximal to the budding domain)? Is it the phenotype of double KO?

5)     Figure 5-7 provided very nice results; however, the proposed model is not easy to follow. It is helpful for readers if authors can provide a cartoon picture of the proposed model.

Author Response

The title should be “Yeast kinesin-5, Cin8, …”. The study focuses on Cin8 and it is unclear Kip1 contributions.

Response: We have changed the title to reflect this. (Page 1).

Although the image of CEN4-GFP in Fig 3B looked good, Figures 3F and 6A were very poor images and difficult to judge the integrity of following quantification.

Response: To better distinguish the cells, we marked the cells of cin8-AID mad1D (Fig. 3F) and cin8-AID dam1-3A (Fig. 6A) with a white dotted line that show chromosome missegregation. We also clarified in the text that cin8-AID mutants do not separate SPBs fully as WT cells because the absence of Cin8 impairs spindle elongation and SPB segregation. However, cin8 mutants are still able to segregate chromosomes into two daughter cells. See Fig. 3F (page 10), Fig. 6A (page 16), and text (page 12).

The use of CEN4-GFP is reasonable but the authors also should perform quantifications of entire kinetochores (such as Mtw1-GFP) and other specific CEN-GFP in addition to CEN4 to support author’s concoctions.

Response: We chose not to use kinetochore protein Mtw1-GFP to perform quantifications because low frequency of chromosome missegregation would be undetectable due to the clustering nature of the 16 KTs in budding yeast; this reasoning is explained in the text. (Page 11). We intended to repeat Fig. 3B/C using GFP-marked centromere of chromosome V (CEN5) and we have used this strain to examine kinetochore clustering (Richmond D. 2013 MBoC). However, all Cin8-AID strains require Tir1 F-box protein to induce degradation of AID-tagged Cin8 by auxin. Tir1-myc has been inserted at the URA3 locus on chromosome V in budding yeast, which is close to CEN5.  We tried tetrad dissection to obtain CEN5-GFP cin8-AID Tir1-Myc strains for this experiment, but were unable to obtain the strains  likely due to the low chance of recombination between CEN5 and Tir1-Myc-URA3. However, we have used CEN4-GFP strains many times to examine chromosome missegregation, and the results are always consistent with the viability loss (Jin F. et al. 2017 Genetics; Bokros M. et al. 2016, Cell Reports; Jin F. et al. 2012 PLoS Genetics). Therefore, we reason that the results using CEN4-GFP strains are reliable to reflect the rate of chromosome missegregation.

Alternatively, we can use chromosome fragment (CFIII) loss protocol (coloney sectoring) developed by Phil Hieter to quantify missegregation in these strains. If you believe it is necessary, we will be happy to perform this experiment, but extra time is needed to construct strains for this experiment.

In Figure 4D bottom cin8-AID/Kip1KO, why weren’t kinetochore clusters retained at the mid zone (proximal to the budding domain)? Is it the phenotype of double KO?

Response: We agree that this phenotype is interesting. After reading the literature, we found a previous report showing that loss of Cin8 causes unbalanced number of astral microtubules emanating from the two SPBs, which is accompanied by an abnormal nuclear positioning (de Gramont et al., cell cycle, 2007). Kip1 had a less significant but present defect, so it is possible that in our cin8-AID kip1D mutant, the additive effects on astral MTs lead to a dramatic nuclear positioning defect. We cited this finding in the text. (Page 13, 14).

Figure 5-7 provided very nice results; however, the proposed model is not easy to follow. It is helpful for readers if authors can provide a cartoon picture of the proposed model. (Reviewer 1)

Response: We agree that a schematic would provide clarification on our proposed model for readers. We included this as Fig. 7E and have referenced it in the text appropriately. (Page 17, 18, and 20).

Reviewer 2 Report

The authors aimed to examine the functional role of kinesin5-like proteins cin8 and kip1 in the checkpoint of yeast.  The experiments were well conducted.  However, I raised only one comment.  

For general readers, the author should show the cartoon how the checkpoint and kinesin5-like proteins work for chromosome division.  

Author Response

For general readers, the author should show the cartoon how the checkpoint and kinesin5-like proteins work for chromosome division. (Reviewer 2).

Response: The other reviewer has the similar opinion. In the revised version, we included a model in Figure 7. We also discuss this model in the text appropriately. (Page 17, 18, and 20).

Reviewer 3 Report

The manuscript by Sherwin et al probed the role of kinesin-5 motor proteins Cin8 and Kip1 in chromosome segregation. They created a conditional mutant of Cin8 by the degron system. The cin8-AID and SAC component protein mad1∆ mutants showed altered chromosome segregation. Interestingly SAC mutation in cin8kip1 suppresses the metaphase arrest. Further, they showed that tension checkpoint mutant dam1-3A causes chromosome missegregation in cin8 depleted cells. Overall, the data support the conclusions very well and I have only a few suggestions for improving the manuscript.

1. The introduction section is too descriptive and some of the information is again repeated in the result section. It would be nice to have focused information based on the manuscript.

2. Figure1A: It looks like the cin8-AID-flag is more stable in the cin8kip1 strain in comparison to the cin8 strain?

3. Fig2: It would be more appropriate to move Fig2B, C as Fig2A, B to follow the text.

4. It’s not clear what’s the difference between Fig2A and 2D?

5. Is it possible to check the localization of Ipl1 in the Cin8-AID mutant?

Author Response

  1. The introduction section is too descriptive and some of the information is again repeated in the result section. It would be nice to have focused information based on the manuscript.

Response: We agree that the introduction was too detailed so we reworded it to focus on key data and outline our proposed model based on our results. (Page 5).

  1. Figure1A: It looks like the cin8-AID-flag is more stable in the cin8 kip1 strain in comparison to the cin8 strain?

Response: We believe this is due to the variance between different experiments or some strain difference. We repeated the experiment to confirm this, and replaced the old western blot with this new, although similar result (Fig. 1A). Even though the auxin-induced degradation of Cin8-AID is incomplete, it is sufficient to stop the cell cycle progression in kip1∆ background.

  1. Fig 2: It would be more appropriate to move Fig2B, C as Fig2A, B to follow the text.

Response: We agree rearranging the text to reflect the order of experiments in Figure 2 makes the manuscript more understandable. We edited the text accordingly. (Page 7).

  1. It’s not clear what’s the difference between Fig2A and 2D?

Response: Fig. 2A and Fig. 2D both are counts of budding index throughout the cell cycle. However, Fig 2A corresponds with the Pds1 levels shown in the western blotting (Fig. 2B), whereas Fig. 2D represents the budding index used for the CEN4-GFP segregation experiment (Fig. 2E). Therefore, they are different strains. We attempted to clarify this in the text in the revised version. (Page 9).

  1. Is it possible to check the localization of Ipl1 in the Cin8-AID mutant?

Response: Previous studies show the centromere/kinetochore localization of CPC in budding yeast prior to anaphase entry, but anaphase entry triggers CPC translocation to the spindle and spindle midzone. Because Cin8 depletion delays anaphase entry, we speculate centromere/kinetochore localization of Ipl1 in metaphase cells of cin8-AID. We did not perform this experiment because it is not clear to us if the result from this experiment will help us to further decipher the role of Cin8 in tension generation or chromosome segregation.

Round 2

Reviewer 1 Report

This revised manuscript was well written and easy to follow. These findings provided valuable information in the field, and I now recommended it for publication in the leading journal, Cells,